# Motivation, Intention and Opportunity: Wearing Masks and the Spread of COVID-19

**Geoff Kaine** [1,*] and **Vic Wright** [2]

1 Manaaki Whenua Landcare Research, Hamilton 3216, New Zealand
2 UNE Business School, University of New England, Armidale 2351, Australia
* Correspondence: kaineg@landcareresearch.co.nz; Tel.: +64-7859-3763

**Abstract:** Prior to the emergence of the Omicron variant, we found large regional differences ($\eta^2 = 0.19$) in the frequency of wearing face masks in New Zealand even though the strength of people's motivation to wear face masks to prevent the spread of COVID-19 was similar across regions. These differences were associated with regional differences (as measured by case numbers) in the risk of COVID-19 infection. The emergence of Omicron and its spread throughout New Zealand in conjunction with the cessation of lockdowns offered the opportunity to test whether regional differences in the frequency of mask wearing disappeared once the risk of COVID-19 infection became uniform across the country. It also created an opportunity to investigate differences in people's behaviour with respect to wearing masks in private and in public. The results confirmed that regional differences in the frequency of mask wearing disappeared once Omicron spread through the country. We also found that the frequency of wearing masks was significantly lower when with family or friends than when out in public.

**Keywords:** behaviour; behavioural intentions; COVID-19; mask wearing; health policy

## 1. Introduction

The success of measures advocated by governments to slow or stop the spread of COVID-19 (coronavirus disease of 2019) depended, in the first instance, on the willingness of individuals to comply with them [1–5]. However, intentions do not necessarily translate into action. Success in translating intentions into actions and changing behaviour also requires that individuals perceive the need to act in a timely fashion. Hence, understanding why individuals may not change their behaviour, despite intending to do so, is critical if policies to encourage measures requiring behaviour change are to be effective.

For example, governments have advocated wearing face masks with a view to slowing the spread of COVID-19, thereby avoiding higher rates of infection and mortality, reducing demands on health and transport systems, and reducing economic [6] and psychological damage [7]. Drawing on the social psychological concepts of involvement and attitudes Kaine et al. [8] found significant and substantial regional differences in the frequency of wearing face masks in public. This was despite finding highly similar patterns across regions in (1) beliefs about COVID-19, (2) the perceived advantages and disadvantages of wearing face masks, (3) attitudes towards wearing face masks and preventing the spread of COVID, and (4) motivation with respect to wearing face masks and preventing the spread of COVID-19 [8].

Kaine et al. [8] argued that if masks are readily available, socially acceptable, and reasonably comfortable, then differences in the frequency of wearing face masks will, arguably must, arise out of differences in perceived need; that is, differences in perceptions of the imminent threat of airborne infection. If this threat is perceived to be high, then an intention to wear a face mask is activated and translates into action (because there is a perceived need). If the threat is perceived to be low, then an intention to wear a face mask

is not activated (because there is no perceived need) and so does not translate into action. Consistent with this argument they found a significant, positive association between the frequency of wearing face masks in public and the risk of exposure to infection based on COVID-19 case numbers in each region [8].

If the argument proposed by Kaine et al. [8] is correct, and if the regional risk of infection from COVID-19 were uniform and beliefs, attitudes, and motivations remained similar across regions, regional differences in the frequency of mask wearing should disappear. The emergence of the Omicron variant in 2021 and its rapid spread throughout New Zealand, in conjunction with the cessation of lockdowns, offered the opportunity to test whether this was, indeed, the case.

Consequently, in this paper we investigate whether regional differences in the frequency of mask wearing across New Zealand disappeared following the emergence of Omicron and its spread across the country. The easing and eventual cessation of lockdowns in 2021, in conjunction with the spread of Omicron across the country, resulted in a situation where people were able to have close contact with friends and acquaintances, provided they wore face masks. This created an opportunity for us to investigate differences in people's behaviour with respect to wearing face masks in public, at work, and in the company of friends.

## 2. Background

### 2.1. Theory

The theory on which this paper is based has been described in detail previously [8–11]. Briefly, the extended decision-making process that directs the non-routine actions of individuals recognises there are two phases to the process: decision and implementation. Extended decision-making processes are triggered when, for example, individuals experience novel situations (such as a pandemic) that require them to consider changing routine behaviours [8]. The natural point of separation between the two phases is the 'action intention' which arises once a decision is made. This is normally referred to as 'behavioural intention' [12–15]. This intention is the new action or actions, such as wearing a face mask, that the individual intends to undertake to meet a triggered, personal aspiration such as avoiding COVID-19 infection [8].

Having formed a behavioural intention, the second stage of the process, decision implementation, comes into play. Decision implementation is routine and familiar to all users when it comes to existing practices and products [8]. In the case of novel practices and products, decision implementation assumes greater importance because it defines the rate of adoption of the novel behaviour or product. Measures such as wearing face masks that were introduced to contain the spread of COVID-19 fit into this novel category: when introduced, this practice was obviously new to most people in New Zealand [8].

The core underlying assumption of models of extended decision making [12,13] is that the behaviour to be explained or predicted is purposive, rather than random, and the product of several sets of inputs: the individual's perception of relevant reality, their general and specific behavioural predispositions related to the behaviour(s) of interest, and the incentive they perceive to allocate scarce cognitive effort to related decisions. At any point in time, these inputs will tend to be correlated because it is known that inconsistency among them is psychologically discomforting, generating cognitive dissonance.

The inputs play different roles, however. Perceived reality can be assessed by investigating salient beliefs of subjects. Beliefs are foundational: deliberate, purposive behaviour must conform with them if dissonance is to be avoided. Behavioural predispositions can be assessed by exploring attitudes (which include values) and opinions. Plainly, coincidence among beliefs, attitudes, and opinions is often likely. Significantly, to change behavioural predispositions and action decisions, changes need to be evoked in beliefs or, less readily, values.

Of greatest importance, however, is the presence or absence of extended decision making: the extent to which the subject turns their attention to the behaviour of interest

to the analyst. The greater this involvement with the decision, the more cognitive effort will be committed, including the possible search for new information about reality, and the greater the potential influence of third parties.

When involvement is low, relevant behaviour, if it exists, may seem random and insensitive to changes in reality that others perceive to be relevant. To the actor, the behaviour may not trigger dissonance because it is of so little personal consequence, being so distant from personal identity, that inconsistencies across relevant beliefs, values, and opinions may not even be sensed.

Bagozzi [12], one of the few theorists to model the implementation of behavioural intentions, draws attention to the fact that different sets of factors can influence the formation of behavioural intentions and their implementation. Hence, while one set of factors influences the creation of an intention, another (possibly overlapping) set may influence the implementation of the intention. In the absence of any barriers to implementation, the most likely explanation for a failure to act will be the absence of a perceived need to act [8]. Given that barriers to use are absent, in that masks are readily available, socially acceptable, and reasonably comfortable, perceived need will relate to the perceived imminent threat of airborne infection.

People's perception of the threat of airborne infection is subjective and will be cue-driven [8]. The cues they employ may well be influenced by reported infections in an area, trends in reported infections, social discussion about them, and perhaps by the prevalence of mask wearing [8]. The adoption of behaviours such as the wearing of face masks has been associated with perceptions of the perceived risk of infection, the local incidence rate of COVID-19, and feelings of stress in relation to COVID-19 [16].

In the next section we provide a brief description of the history of COVID-19 in New Zealand to place the subsequent analysis in its proper context.

*2.2. COVID-19 in New Zealand*

Following initial detection of COVID-19 in New Zealand in early 2020, the central government closed New Zealand's international border to all except returning citizens and permanent residents and instituted a four-tier alert system [17]. This system mandated policy measures such as: progressively tighter restrictions on people's movement beyond their homes and immediate families, including travelling to work; social distancing and encouraging the wearing of masks outside the home at the higher alert levels; and self-isolating and seeking testing if people felt unwell or experienced symptoms characteristic of COVID-19 infection [17].

A National State of Emergency was declared in March 2020 and a Level 4 'lockdown', the highest level of alert, implemented [17]. The country progressively moved to lower alert levels: Level 3 towards the end of April and Level 2 in early May 2020 as the spread of the virus slowed and stopped [8]. The lowest level, Alert Level 1, was introduced in June 2020 because community transmission had halted and there were no active cases in the country outside the Managed Isolation and Quarantine facilities (MIQ) [8]. These facilities were established specifically to confine all travellers to New Zealand for 14 days after arrival [8]. If a traveller tested positive for COVID-19 at any time during the 14 days, they were moved to another quarantine facility for people with COVID-19 [17]. Over the following 18 months, Alert levels varied regionally depending on the detection of COVID-19 cases (see [17] for details).

The central government commenced a mass vaccination programme for COVID-19 using the Pfizer vaccine, starting with border staff and MIQ workers, in February 2021 [11,18]. The programme was accompanied by an extensive, government-funded publicity campaign using traditional and social media. By the end of 2021, over 90% of eligible New Zealanders had received two doses of the Pfizer vaccine [19].

In December 2021, following the spread of the Omicron variant through New Zealand, the COVID-19 Protection Framework was introduced [20]. The Framework was intended to:

- help people protect one another from the virus,

- keep hospitalisation rates as low as possible and avoid overwhelming the health system,
- minimise the impact of large outbreaks,
- reduce the need for lockdowns, and
- give people and businesses more stability [20].

Under the Framework, travel was permitted throughout New Zealand, and all workplaces and schools reopened. The Framework had red, orange and green settings. At red and orange settings, face masks were to be worn when indoors (apart from the home). There were also social-distancing requirements and capacity limits at some venues under the red setting. All restrictions, including the requirement to wear masks indoors, were removed at the green setting [20].

Several regions in the North Island were placed at the red setting when the Framework was implemented in December 2021, the remainder of the country being at the orange setting. In January 2022, the first cases of community transmission of Omicron were confirmed and all New Zealand was moved to the red setting. In the same month, the central government instituted a three-phase health response to slow the spread of Omicron. Phase three, which was initiated in late February 2022 when case numbers were in the thousands, largely relied on self-testing for COVID-19 and self-isolating along with household contacts [21].

In addition, in early February requirements regarding the wearing of face masks were revised. Face masks were made compulsory for school children (and their teachers) in Year 4 and above and everyone was required to wear masks in close-proximity businesses, food and drink businesses, and at events and gatherings (though the mask could be removed to eat and drink) [22]. People in non-public-facing workplaces were encouraged to wear masks. By early April 2022 most vaccine mandates had been abandoned. The COVID-19 Protection Framework was abandoned in September 2022 [20].

## 3. Materials and Methods

### 3.1. Survey Data

A national survey was conducted of New Zealand residents during March 2022 following the detection of community transmission of the Omicron variant. When the survey commenced, all residents were under the red setting of the COVID-19 Protection Framework, which meant that they were at least expected, if not required, to wear masks in all public places [20].

A questionnaire was designed based on the $I_3$ Compliance Framework [8,9]. The questionnaire replicated much of the content of Kaine et al. [8,10] in relation to wearing face masks and sought information from the public on their beliefs about, attitudes towards, and willingness to slow the spread of COVID-19 and to wear face masks. It also contained questions, based on Kaine et al. [11], on beliefs regarding COVID-19 vaccines. Involvement was measured using a condensed version of the Laurent and Kapferer [23] involvement scale (described in Kaine [8]), with respondents rating two statements on each of the five components of involvement (functional, experiential, identity-based, risk-based, and consequence-based). Attitudes were measured using a simple, evaluative scale (the questionnaire is reproduced in Table S1). The ordering of the statements in the involvement and attitude scales was randomised to avoid bias in responses [8]. Following [8], respondents indicated their agreement with statements in all the involvement, attitude, and belief scales using a five-point rating, ranging from strongly disagree (1) to strongly agree (5).

Respondents' propensity to wear face masks was obtained by asking them if they had worn a face mask in five social settings; namely when:

(i)     they were out in public the previous week,
(ii)    they had gone out to work the previous week,
(iii)   they visited friends at their home,
(iv)    they had friends visit them at their home, or
(v)     they exercised outside.

Respondents answered using a five-point scale ranging from 'always' to 'never'.

Following Kaine et al. [8], information was also sought on the demographic characteristics of respondents, including age, education, and ethnicity, and whether they wore masks. The ethnicity categories were Māori (the Indigenous people of New Zealand), European New Zealander, Pacific Islander, Asian, and Other [8,10].

As with Kaine et al. [8,10], participation in surveys was voluntary, respondents could leave the survey at any time, and all survey questions were optional and could be skipped. The research approach was reviewed and approved by the Manaaki Whenua–Landcare Research's social ethics process (application no. 2021/10 NK) which is based on the New Zealand Association of Social Science Research code of ethics [8].

The questionnaire was completed by a large random sample of residents ($n = 1000$), stratified by regional population, who were members of a large-scale, commercial consumer internet panel. The sample size ensured there was a 95% chance that sample values were within 3% of population values. Panel members received reward points (which are redeemable for products and services) for completing surveys [8]. An internet link to the questionnaire was distributed to randomly selected members of the panel subject to the constraint that they were resident in the relevant region and were not minors [8].

Data were also gathered on the number, dates, and location by District Health Board of COVID-19 cases reported by the New Zealand Ministry of Health [24]. At the time of the survey there had been more than 120,000 cases of COVID-19 throughout New Zealand [24]. The data is reproduced in Table S2.

*3.2. Methods*

Involvement scores were computed for each respondent as the simple arithmetic average of their agreement ratings for the 10 statements in the involvement scales [8]. Attitude scores were computed as the simple arithmetic average of their agreement ratings for the five statements in the attitude scales [8].

The analysis was split into several steps as follows. First, to avoid problems with multicollinearity among the belief variables, factor analysis was employed using the raw data on beliefs to create composite, uncorrelated belief variables for use in subsequent regression analyses [25]. This was achieved using principal component analysis with varimax rotation. The threshold for the number of factors was set at an eigenvalue of 1. The results from the factor analyses for beliefs about COVID-19 and about COVID-19 vaccines, and beliefs about the advantages and disadvantages of wearing face masks, are reported in Section 4.2.

We then investigated the associations between beliefs and attitudes, and the associations between beliefs and involvement, by estimating the following regressions:

$$\text{ICOVID} = b_0 + b_i \sum \text{FC}_i \tag{1}$$

$$\text{ACOVID} = b_0 + b_i \sum \text{FC}_i \tag{2}$$

$$\text{IMASK} = b_0 + b_i \sum \text{FC}_i + b_j \sum \text{FM}_j \tag{3}$$

$$\text{AMASK} = b_0 + b_i \sum \text{FC}_i + b_j \sum \text{FM}_j \tag{4}$$

where ICOVID and ACOVID are involvement with, and attitude towards, slowing the spread of COVID-19, respectively. IMASK and AMASK are involvement with, and attitude towards, wearing face masks, respectively. $\text{FC}_i$ and $\text{FM}_j$ are the sets of composite variables resulting from the principal component analysis of beliefs about COVID-19 and about COVID-19 vaccines, and beliefs about the advantages and disadvantages of wearing face masks, respectively.

We also investigated the consistency of beliefs about the benefits of slowing the spread of COVID-19 and beliefs about COVID-19 and COVID-19 vaccines by estimating the following regressions:

$$BSPREAD = b_0 + b_i \sum FC_i \tag{5}$$

where BSPREAD is one of the set of beliefs about the benefits of slowing the spread of COVID-19. The results of these regression analyses are also reported in Section 4.2.

Following Kaine et al. [8,10], we hypothesised that respondents' intentions in terms of willingness to take some responsibility for slowing the spread of COVID-19, and their willingness to change normal behaviour, work with others, and make sacrifices to slow the spread of COVID-19 would be a function of their involvement with, and attitude towards, slowing the spread of COVID-19 in New Zealand. To test these hypotheses we estimated the following regressions:

$$INTENT = b_0 + b_1 \, ICOVID + b_2 \, ACOVID + b_k \sum X_k \tag{6}$$

where INTENT is one of the set of intentions variables, ICOVD is involvement and ACOVID attitude, and $X_k$ are respondents' demographic characteristics which were included to account for the possibility that the demographic differences among respondents might be correlated with relevant omitted variables (e.g., social norms, susceptibility to infection, risk of severe symptoms). The results of these regression analyses are reported in Section 4.3.

Again following Kaine et al. [8,9], we hypothesised that respondents' propensity to wear face masks would be a function of their involvement (IMASK) with, and attitude (AMASK) towards, wearing face masks. Consequently, we estimated the following regressions:

$$MASK = b_0 + b_1 \, IMASK + b_2 \, AMASK + b_k \sum X_k \tag{7}$$

where MASK is the self-reported frequency of wearing a face mask in each of the five social settings described previously. Again, respondents' demographic characteristics were included to account for the possibility that the demographic differences among respondents might be correlated with relevant omitted variables (e.g., social norms, susceptibility to infection, risk of severe symptoms). The results of these regression analyses are reported in Section 4.4.

Lastly, we expected regional differences in the frequency of mask wearing across New Zealand to disappear following the emergence of Omicron and its spread across the country. We tested this hypothesis in two ways. First, following Kaine et al. [8], we included explanatory variables in the regressions where we assumed that respondents' perception of the risk of infection was either proportional to the total number of COVID-19 cases reported in their region prior to the survey, or to the total number of cases expressed as a fraction of the population of the region [8]. That is:

$$MASK = b_0 + b_1 \, IMASK + b_2 \, AMASK + b_k \sum X_k + b_m \sum INCIDENCE_m \tag{8}$$

$$MASK = b_0 + b_1 \, IMASK + b_2 \, AMASK + b_k \sum X_k + b_m \sum CASES_m \tag{9}$$

where $CASES_m$ and $INCIDENCE_m$ are the total number of COVID-19 cases reported in each region prior to the survey and the total number of cases in each region expressed as a fraction of the population of the region, respectively.

Second, we estimated regressions with the addition of regional dummy variables to account for any regional differences in the frequency of mask wearing.

$$MASK = b_0 + b_1 \, IMASK + b_2 \, AMASK + b_k \sum X_k + b_m \sum DUMMY_m \tag{10}$$

where $DUMMY_m$ are regional dummy variables. The results of these regression analyses are described in Section 4.5 and reported in detail in Appendix A.

Statistical analyses were conducted using IBM SPSS Statistics v28, Windows [26]. Given the size of the sample, we set the level of statistical significance at *p* < 0.01 to ensure we only interpreted associations that were both statistically significant and meaningful in terms of effect size [27].

## 4. Results

To begin with, the survey sample for this study was broadly similar to that analysed by Kaine et al. [8,10] with regard to the distribution of respondents' age, education, income composition, gender, and ethnicity (see Table 1). Approximately 43% of respondents to the survey were women.

**Table 1.** Age, education, ethnicity, and income distribution of respondents.

| Age Category (Years) | Kaine et al. [9] Respondents | Sample for This Study | Kaine et al. [8] Respondents |
|---|---|---|---|
| 18–29 | 13.2 | 7.9 | 22.8 |
| 30–39 | 22.6 | 28.4 | 21.8 |
| 40–49 | 21.5 | 17.7 | 18.4 |
| 50–59 | 12.5 | 14.2 | 13.1 |
| 60–69 | 13.8 | 12.5 | 12.5 |
| 70 and over | 16.4 | 19.2 | 11.4 |
| **Education category** | | | |
| Some or all of secondary school | 19.5 | 18.2 | 14.2 |
| Certificate (1–6) | 19.1 | 15.8 | 12.4 |
| Diploma (5–7) | 17.5 | 14.5 | 14.3 |
| Bachelor | 23.4 | 24.8 | 33.6 |
| Post-graduate diploma/certificate | 11.1 | 11.8 | 10.2 |
| Post-graduate degree | 9.4 | 14.9 | 15.3 |
| **Ethnic category** | | | |
| European | 72.1 | 76.0 | 53.3 |
| Māori | 13.5 | 5.2 | 4.4 |
| Pacific Islander | 1.8 | 2.3 | 4.7 |
| Asian | - | 14.5 | - |
| Other | 12.7 | 2.0 | 37.6 |
| **Income category** | | | |
| Less than $20,000 | 8.5 | 3.8 | 4.3 |
| $20,000 to $50,000 | 26.0 | 19.3 | 21.2 |
| $50,000 to $70,000 | 21.7 | 18.0 | 18.6 |
| $70,000 to $100,000 | 22.1 | 19.9 | 22.0 |
| More than $100,000 | 21.6 | 39.1 | 33.8 |

Values are proportions in each sample.

We conducted a reliability analysis [28] to check that the scales for measuring attitudes and involvement were acceptably robust. Overall, the results indicate that the reliability of the scales was acceptable with values for Cronbach's alpha of 0.875 and 0.874, respectively, for involvement with slowing the spread of COVID-19 and with wearing face masks, and 0.930 for attitude towards wearing face masks.

### 4.1. Factor Analysis

Beliefs about COVID and COVID vaccinations, when condensed into 4 composite factors, accounted for 60% of the variance in the set of 14 relevant belief variables (see Table 2). We interpreted the first factor as describing beliefs that the threat of infection from COVID-19 is limited, the second as describing a perception that the threat to personal health is exaggerated, the third as describing the benefits of vaccination, and the fourth as

describing beliefs about modes of infection. We notate these factors as FC1, FC2, FC3 and FC4, respectively.

Respondents' beliefs about wearing masks were condensed into two composite factors which accounted for 56% of the variance in the set of 14 belief variables (Table 3). We interpreted the first factor as representing beliefs that face masks are ineffective and impractical and the second as representing beliefs that the face masks available to the public are of poor quality and they are poorly used. We notate these factors as FM1 and FM2, respectively.

**Table 2.** Correlation between factors and beliefs about COVID-19 and COVID-19 vaccinations.

| Belief | FC1–Threat of Infection Is Limited | FC2–Threat to Health Is Exaggerated | FC3–Vaccination Effects | FC4–Modes of Infection |
|---|---|---|---|---|
| Once you are vaccinated you cannot catch or spread COVID-19 | 0.79 | | | |
| Children cannot catch COVID-19 | 0.78 | | | |
| You cannot catch COVID-19 from people with the virus who do not have symptoms | 0.64 | | | |
| Children are perfectly safe from COVID-19 | 0.64 | 0.38 | | |
| I think COVID-19 is a hoax | 0.57 | 0.42 | | |
| Once you have had COVID-19 you are immune to re-infection | 0.57 | | | |
| Fears about COVID-19 are exaggerated | | 0.77 | | |
| COVID-19 is no worse than the seasonal flu | 0.34 | 0.69 | | |
| COVID-19 is only a danger to the elderly and people who already have health problems | 0.32 | 0.67 | | |
| COVID-19 is a real threat to my health | | −0.66 | | 0.35 |
| Getting vaccinated means you will recover faster | | | 0.84 | |
| Getting vaccinated means your symptoms will be weaker | | | 0.83 | |
| You can catch COVID-19 by touching anything handled by an infected person | | | | 0.82 |
| Infected people spread COVID-19 by coughing and sneezing | −0.30 | | | 0.55 |

### 4.2. Beliefs, Involvement, and Attitudes

The purpose of this analysis was to confirm that involvement and attitudes are strongly, and plausibly, associated with beliefs. The estimates for Equations (1) and (2) show that beliefs about COVID-19 and COVID-19 vaccines explained more than 40% of the variance in attitudes and involvement regarding slowing the spread of COVID-19 (Table 4). Similarly, the estimates for Equations (3) and (4) show that beliefs about the advantages and disadvantages of wearing face masks explained around 60% or more of the variance in involvement with, and attitudes towards, wearing face masks (see Table 4). Note that for these and all subsequent regressions, the F-test $p$-value is less than 0.001.

**Table 3.** Correlation between factors and beliefs about wearing face masks.

| Belief | FM1–Ineffectiveness and Impracticality of Masks | FM2–Poor Quality and Use |
|---|---|---|
| Wearing a face mask sets a good example to others | −0.85 | |
| Wearing face masks should be compulsory | −0.84 | |
| Face masks are effective in preventing the spread of COVID-19 | −0.80 | |
| Wearing face masks to slow the spread of COVID-19 is just not practical | 0.71 | |
| People who wear face masks are over-reacting | 0.71 | |
| You should only have to wear a face mask if you feel unwell | 0.69 | |
| You should only have to wear a face mask if you are old or have a health problem | 0.60 | |
| Face masks are too difficult and inconvenient to wear if you have glasses | 0.48 | |
| Face masks are just too uncomfortable | 0.48 | |
| Face masks are not much help unless you wear gloves as well | | 0.65 |
| Face masks are not much help in slowing the spread of COVID-19 because people do not wear them properly | | 0.64 |
| Home-made face masks are a waste of time and effort | | 0.60 |
| The kind of face masks we can buy are not worth bothering with | | 0.57 |
| Face masks on their own are not much help in slowing the spread of COVID-19 | | 0.54 |

**Table 4.** Standardised parameter estimates for involvement and attitudes towards preventing the spread of COVID-19 and wearing face masks.

| Composite Belief | Involvement with Slowing the Spread of COVID-19 | Attitude towards Slowing the Spread of COVID-19 | Involvement with Wearing Face Masks | Attitude towards Wearing Face Masks |
|---|---|---|---|---|
| FC1–Threat of infection is limited | −0.078 ($p < 0.001$) | −0.161 ($p < 0.001$) | 0.155 ($p < 0.001$) | −0.062 ($p < 0.001$) |
| FC2–Threat to health exaggerated | −0.552 ($p < 0.001$) | −0.527 ($p < 0.001$) | −0.173 ($p < 0.001$) | −0.086 ($p < 0.001$) |
| FC3–Vaccination effects | 0.301 ($p < 0.001$) | 0.297 ($p < 0.001$) | 0.100 ($p < 0.001$) | 0.090 ($p < 0.001$) |
| FC4–Mode of COVID-19 infection | 0.175 ($p < 0.001$) | 0.152 ($p < 0.001$) | 0.088 ($p < 0.001$) | 0.067 ($p < 0.001$) |
| FM1–Ineffectiveness and impracticality of masks | | | −0.610 ($p < 0.001$) | −0.734 ($p < 0.001$) |
| FM2–Poor quality and use | | | −0.081 ($p < 0.001$) | −0.236 ($p < 0.001$) |
| Adjusted $R^2$ | 0.43 | 0.41 | 0.58 | 0.78 |

Number of observations was 1000 for each regression.

Involvement with slowing the spread of COVID-19 decreases with the belief that the threat of infection is limited and the threat to health is exaggerated. Involvement increases with the belief that vaccination is beneficial and that surfaces, as well as aerosols, are a mode of infection. Believing that the threat of infection is limited and the threat to health is exaggerated has an unfavourable effect on attitude towards slowing the spread of COVID-19, while believing that vaccination is beneficial and that surfaces as well as

aerosols are a mode of infection has a favourable effect on attitude towards slowing the spread of COVID-19.

Involvement with wearing masks increases with the belief that the threat of infection is limited (to adults with symptoms), that vaccination is beneficial, and that surfaces as well as aerosols are a mode of infection. Involvement decreases with the belief that the threat to personal health from COVID-19 is exaggerated. Involvement with face masks also decreases if masks are believed to be impractical, ineffective, poor quality, and poorly used. Believing that the threat of infection is limited, and that the threat to personal health from COVID-19 is exaggerated, has an unfavourable effect on attitude towards wearing masks. Believing that vaccination is beneficial, and that surfaces as well as aerosols are a mode of COVID-19 infection, has a positive effect on attitude. Believing that masks are impractical, ineffective, poor quality, and poorly used has an unfavourable effect on attitude towards wearing masks.

The estimates for the regressions described by Equation (5) show that beliefs about COVID-19 and COVID-19 vaccines also explained a substantial proportion, 30% or more, of participants' beliefs about the consequences of attempting to slow the spread of COVID-19 (see Table 5).

**Table 5.** Standardised parameter estimates for beliefs about preventing the spread of COVID-19.

| | Stopping the Spread of COVID-19 Saves Lives | Slowing the Spread of COVID-19 Will Keep Supply Chains Functioning | Better to Let COVID-19 Spread and Build Herd Immunity | There Is No Point Trying to Stop the Spread as It Is a Virus and Will Keep Changing | Slowing the Spread of COVID-19 Will Reduce the Pressure on Our Health System |
|---|---|---|---|---|---|
| FC1–Threat of infection is limited | −0.113 ($p < 0.001$) | −0.132 ($p < 0.001$) | 0.229 ($p < 0.001$) | 0.232 ($p < 0.001$) | −0.198 ($p < 0.001$) |
| FC2–Threat to health exaggerated | −0.512 ($p < 0.001$) | −0.379 ($p < 0.001$) | 0.537 ($p < 0.001$) | 0.472 ($p < 0.001$) | −0.459 ($p < 0.001$) |
| FC3–Vaccination effects | 0.296 ($p < 0.001$) | 0.299 ($p < 0.001$) | −0.157 ($p < 0.001$) | −0.169 ($p < 0.001$) | 0.301 ($p < 0.001$) |
| FC4–Mode of COVID-19 infection | 0.183 ($p < 0.001$) | 0.163 ($p < 0.001$) | | | 0.133 ($p < 0.001$) |
| Adjusted $R^2$ | 0.39 | 0.27 | 0.36 | 0.30 | 0.36 |

Number of observations was 1000 for each regression.

### 4.3. Intentions, Involvement, and Attitudes

The regressions described by Equation (6) that predict respondents' behavioural intentions are reported in Table 6. They indicate that at least 60% of the variation in respondents' willingness to take some responsibility for slowing the spread of COVID-19, and their willingness to change normal behaviour, their willingness to work with others, and their willingness to make sacrifices to slow the spread of COVID-19 was explained by their involvement with, and attitude towards, slowing the spread of COVID-19.

The results show that involvement and attitude account for the bulk of the explained variation in intentions. They also reveal the consistency one would expect across beliefs, attitudes, and intentions. The variation in respondents' intentions was only weakly related to their demographic characteristics.

**Table 6.** Standardised parameter estimates for behavioural intentions.

| | Feel Responsible for Eliminating COVID-19 (*n* = 846) | Prepared to Change Normal Behaviour (*n* = 994) | Willing to Make Sacrifices (*n* = 993) | Willing to Work Together (*n* = 1000) |
|---|---|---|---|---|
| Involvement with preventing spread of COVID-19 | 0.315 (*p* < 0.001) | 0.327 (*p* < 0.001) | 0.305 (*p* < 0.001) | 0.287 (*p* < 0.001) |
| Attitude towards preventing spread of COVID-19 | 0.493 (*p* < 0.001) | 0.513 (*p* < 0.001) | 0.540 (*p* < 0.001) | 0.596 (*p* < 0.001) |
| Gender | 0.086 (*p* < 0.001) | 0.084 (*p* = 0.001) | 0.058 (*p* = 0.003) | |
| Income | 0.071 (*p* = 0.002) | | | |
| Asian | | | −0.055 (*p* = 0.005) | |
| Adjusted R$^2$ | 0.58 | 0.63 | 0.64 | 0.70 |

*4.4. Predicting Behaviour*

The frequency distributions for wearing face masks in the five social settings are reported in Table 7 and reveal significant differences in self-reported behaviour from setting to setting. The estimated regressions for predicting the frequency of mask wearing in different social settings, as described by Equation (7), are reported in Table 8. The frequency with which face masks were worn when out in public or at work depended on involvement with, and attitude towards, face masks. As was the case with behavioural intentions, the variation in respondents' wearing of face masks in public was partly related to their demographic characteristics.

**Table 7.** Frequency distribution of self-reported wearing of face masks.

| Setting | Never | Rarely | Sometimes | Often | Always | Total |
|---|---|---|---|---|---|---|
| Wore a face mask in public | 6.6 | 4.6 | 14.7 | 20.6 | 53.4 | 100 |
| Wore a face mask at work | 11.7 | 5.3 | 12.9 | 14.2 | 56.0 | 100 |
| While exercising outside | 55.8 | 15.3 | 13.4 | 6.5 | 9.1 | 100 |
| Visiting friends at their place | 46.2 | 15.4 | 18.7 | 8.1 | 11.7 | 100 |
| Friends visiting your place | 54.7 | 17.3 | 14.4 | 6.8 | 6.8 | 100 |

Values are percentage of respondents answering.

While we found involvement had a significant and substantial effect in the regressions for the frequency of wearing face masks when exercising or being with friends, attitude was not significant. We hypothesised that, in these settings (being outdoors or being with people with whom respondents were acquainted, respectively), respondents' perceptions of the risk of infection may differ from their perception of the risk of infection when they are in a confined space with strangers. This suggests that the frequency of mask wearing in these settings may be influenced more by specific beliefs about COVID-19 and the advantages and disadvantages of wearing face masks than by a generalised attitude towards wearing face masks.

We re-estimated regressions for the frequency of mask wearing when exercising outside and when with friends. We replaced, as explanatory variables, attitude towards mask wearing with the composite belief variables, generated in the factor analysis, regarding COVID-19, COVID-19 vaccines, and mask wearing. The results revealed that respondents who believed that threat of infection from COVID-19 was limited (to adults with symptoms) were less likely to wear face masks when they were with their friends (or exercising outdoors). It seems they perceive that the risk of being exposed to an infected adult with symptoms is lower when meeting friends than when meeting strangers (when out in public

or at work). This seems reasonable as people are more likely to know whether their friends have COVID-19 or COVID-19 symptoms. People were less likely to wear masks when exercising outdoors the more ineffective and impractical they thought masks were.

**Table 8.** Standardised parameter estimates for wearing face masks.

| | Wore a Face Mask in Public | Wore a Face Mask at Work | While Exercising Outside | Visiting Friends at Their Place | Friends Visiting Your Place |
|---|---|---|---|---|---|
| Involvement with wearing face masks | 0.170 ($p < 0.001$) | 0.126 ($p = 0.009$) | 0.217 ($p < 0.001$) | 0.356 ($p < 0.001$) | 0.340 ($p < 0.001$) |
| Attitude towards face masks | 0.451 ($p < 0.001$) | 0.354 ($p < 0.001$) | | | |
| FC1–Threat of infection is limited | | | 0.179 ($p < 0.001$) | 0.106 ($p < 0.001$) | 0.152 ($p < 0.001$) |
| FM1–Ineffectiveness and impracticality of masks | | | $-0.190$ ($p < 0.001$) | | |
| Age | $-0.079$ ($p = 0.003$) | | | 0.188 ($p < 0.001$) | |
| Gender | 0.125 ($p < 0.001$) | 0.125 ($p < 0.001$) | | | $-0.095$ ($p = 0.006$) |
| Asian | | 0.119 ($p < 0.001$) | 0.161 ($p < 0.001$) | 0.195 ($p < 0.001$) | 0.146 ($p < 0.001$) |
| Income | | | | | $-0.114$ ($p = 0.001$) |
| Adjusted $R^2$ | 0.35 | 0.24 | 0.21 | 0.24 | 0.22 |
| Observations | 985 | 678 | 816 | 813 | 696 |

As was the case with mask wearing in public and at work, the variation in respondents' wearing of face masks when exercising outside or with friends was partly related to their demographic characteristics.

### 4.5. Regional Differences in Mask Wearing Behaviour

Consistent with our hypotheses (Equations (8)–(10)), we did not detect any regional differences in the self-reported frequency of mask wearing across New Zealand. Regional dummy variables, regional COVID-19 case numbers, and regional COVID-19 case incidence were not significant in any of the behavioural regressions (see Appendix A).

## 5. Discussion

As with Kaine et al. [8], our results suggest that, on average, respondents in different regions of New Zealand have similar beliefs about, attitudes towards, and motivations regarding slowing the spread of COVID-19 and wearing face masks. They were also similar, on average, regarding their intentions to take some responsibility for slowing the spread of COVID-19, and their intentions to change their normal behaviour, work with others, and make sacrifices to slow the spread of COVID-19 in New Zealand.

In contrast to Kaine et al. [8], respondents in different regions were, on average, similar with respect to the self-reported frequency of wearing face masks in public and in other social settings following the arrival and spread of Omicron in New Zealand. These findings are consistent with the hypothesis that, if regional difference in self-reported frequency of wearing face masks observed by Kaine et al. [8] were attributable to differences in perception of the risk of infection arising from regional differences in observed cases of COVID-19, these differences would vanish once COVID-19 was widespread across New Zealand.

While we found involvement had a significant and substantial effect on the frequency of wearing face masks when out in public, at work, mixing with friends, and exercising

outdoors, participants' general attitude toward wearing face masks only influenced the frequency of wearing face masks in settings where close contact with strangers was likely (out in public or at work). We found that the frequency of wearing face masks when with friends, or exercising outdoors, was influenced by specific beliefs about COVID-19 and the advantages and disadvantages of wearing face masks. We found that respondents who believed that the threat of infection from COVID-19 was limited to adults with symptoms were less likely to wear face masks when they were with their friends or exercising outdoors. This suggests that in these settings, respondents' perceptions of the risk of infection may differ when with people with whom they are acquainted, or when exercising outdoors where encounters with strangers are likely to be fleeting, from their perception of the risk of infection when they are in a confined space with strangers.

### 5.1. Implications

Many studies have investigated people's intentions to change their behaviours to slow the spread of COVID-19 [29–36]. They have found that intentions to change behaviour depend on people's beliefs about, and attitudes towards, those behaviours. Consequently, these studies recommend investing in promotion to change beliefs about, and so attitudes towards, preventative behaviours to increase their adoption. As follows, our results have three important implications for such recommendations.

The first is that our findings reinforce the Kaine et al. [8] reminder that intentions [37–40] do not always immediately, or inevitably, translate into actions. Changing beliefs and attitudes can change behaviour; however, promotional efforts seeking to change preventative health behaviours by changing beliefs and attitudes are unlikely to meet with comprehensive success unless health authorities also seek to identify the factors that:

- trigger the translation of intentions into action, and
- prevent those who are intending to act from acting.

For example, Kaine et al. [8] suggested that respondents to their survey may have relied on the number of COVID-19 infections in their region, together with changes in lockdown levels, as signals to trigger the translation of intention into action with respect to wearing face masks for self-protection [39,41]. This may help to explain location-based differences in the wearing of face masks [42]. Consequently, providing timely and easily accessible location-based information on the number of infections resulting from community transmission is important.

Relatedly, we have found that beliefs about the threat of COVID-19 infection, apparently based on beliefs about potential sources of COVID-19 infection, lead to differences in the frequency with which people wear face masks when out in public compared to when they are with friends. This may help to explain differences in the wearing of face masks in different social settings [43]. It suggests that the public, in New Zealand at least, have views as to the social settings which facilitate community transmission of COVID-19 and vary their behaviour accordingly. Hence, people may not wear face masks when they are with friends because:

- they generally do not wear them (because they have low involvement with slowing the spread of COVID-19); or
- they have high involvement with slowing the spread of COVID-19 but believe the risk of infection is low when they are with friends.

Consequently, providing timely and easily accessible information on the settings that facilitate or mitigate community transmission of COVID-19 is important to the success of prevention measures.

Second, and relatedly, our findings point to the importance of providing accurate, timely and easily understandable information on the danger to health posed by different variants of COVID-19 both in the long term and the short term. This is essential if the public is to:

- set a reasonable criterion for judging the number of infections from community transmission that they should observe to trigger action. This is supposing that there is a relationship between the seriousness of the health risk posed by a variant and the threshold for infections by community transmission below which intentions remain intentions [8]; and
- make reasonable judgements about the severity of the risk of infection in different social settings.

Hence, understanding public perception of the personal risk of COVID−19 infection is fundamental for establishing effective prevention measures [44–46]; for instance, understanding public perceptions of the risks and consequences for personal health of repeated COVID-19 infections (such as the chances of contracting long-COVID and its likely severity), given that Omicron is ubiquitous.

Third, as Kaine et al. [8] observed, it is important to bear in mind that, for New Zealanders at least, wearing a face mask when out in public or with friends means constantly disrupting routine, daily behaviours. Consequently, wearing a face mask requires much more time and effort than, say, being vaccinated for COVID-19, a non-routine action that only needs to be performed a few times. Unfortunately, this suggests that the public would tend to ignore promotional efforts encouraging them to wear masks (and to meet outdoors as much as possible) when with friends once lockdowns ended and travel restrictions were relaxed. This is particularly so if the public perceived infection with the Omicron variant as having only mild consequences, which is likely among those that had been vaccinated. In these circumstances, infections can be expected to spread extremely rapidly and widely, as was the case with Omicron in New Zealand [47]. Hence, understanding public perception of the personal risk of COVID−19 infection in different social settings is critical to judging the timing of lockdowns, especially their termination.

Finally, people's willingness to wear face masks depends on their beliefs about the effectiveness of face masks in protecting them from infection [8,48]. Consequently, developing and promoting to the public clear guidelines on wearing face masks, and increasing promotional efforts dispelling negative myths about the efficacy of masks, are important strategies for encouraging the wearing of face masks [38]. It is important to be aware, though, that people with high involvement may engage in motivated reasoning, i.e., filtering out information that challenges their beliefs and attitudes [49]. If this is the case among those with an unfavourable attitude towards wearing face masks, then promotional efforts dispelling myths about the efficacy of face masks may not change the attitudes of these people.

*5.2. Limitations and Areas for Future Research*

Our findings are subject to several qualifications. First, as the survey sample was drawn from an internet-based consumer panel, there may be selection bias. While the extent of this bias is unknown, it does seem reasonable to suppose that people with low-to-mild involvement may be under-represented in the sample.

Second, social desirability bias [50,51] may have affected self-reporting of the frequency of mask wearing. However, the difference in self-reported frequency of wearing face masks in Kaine et al. [8] suggests that the degree of social desirability bias is likely to be small.

Third, as Kaine et al. [8] observed, the adoption of behaviours such as the wearing of face masks has been associated with a range of variables including feelings of stress in relation to COVID-19 [16]. We did not include such variables in our analysis and, while the correlation between these variables and involvement is unknown, it is likely to be positive. Relatedly, the adoption of preventive behaviours such as the wearing of face masks has been associated with a range of psychological traits such as pro-sociability and empathy [52–54]. The correlation between these traits and involvement deserves further study.

Fourth, the potential for perceptions of the risk of infection to vary across social settings is worth investigating, as are the cues used by the public to infer such risks. While there are numerous studies of risk perception with respect to COVID-19 [44,46,55–57] and numer-

ous studies into the effects of demographic characteristics on mask wearing [42,46,58–61], studies that investigated mask-wearing behaviour in different social settings appear to be uncommon [43,62].

Lastly, the extent to which our results and findings generalise to other countries and epidemics is unknown.

## 6. Conclusions

Governments are seeking to slow the spread of COVID-19 by implementing measures that encourage, or mandate, changes in people's behaviour. The success of these measures depends on people's willingness to change their behaviour and their commitment and capacity to translate that intention into actions. Our findings were consistent with the hypothesis that differences in people's perceptions of the risk of COVID-19 infection result in differences in the frequency with which they wear face masks, even though they have similar beliefs about, and attitudes towards, slowing the spread of COVID-19 and the advantages and disadvantages of wearing face masks.

We also found evidence that people's mask-wearing behaviour varied depending on social settings. It appears that respondents' perceptions of the risk of infection may differ when they are with people with whom they are acquainted or when exercising outdoors (where encounters with strangers are likely to be fleeting) compared to their perception of the risk of infection when they are in a confined space with strangers (out in public or at work).

These results clearly show that intentions do not necessarily translate into actions and that efforts to change behaviour, by seeking to change beliefs and attitudes, can be misplaced if the factors that influence the translation of intentions into action are ignored.

**Supplementary Materials:** The following supporting information can be downloaded at: https://www.mdpi.com/article/10.3390/covid3040043/s1, Table S1: Questionnaire; Table S2: Data.

**Author Contributions:** Conceptualization, G.K. and V.W.; methodology, G.K. and V.W.; formal analysis, G.K.; investigation, G.K. and V.W.; data curation, G.K.; writing—original draft preparation, G.K. and V.W.; writing—review and editing, G.K. and V.W.; project administration, G.K.; funding acquisition, G.K. All authors have read and agreed to the published version of the manuscript.

**Funding:** This research was funded from the Manaaki Whenua Landcare Research Strategic Investment Fund. MWLR project number: PRJ3178.

**Institutional Review Board Statement:** The study was conducted in accordance with the Declaration of Helsinki and approved by the Ethics Committee of Manaaki Whenua–Landcare Research (protocol code 2021/10 NK, 27 January 2021).

**Informed Consent Statement:** Informed consent was obtained from all subjects involved in the study.

**Data Availability Statement:** The data presented in this study are available in Supplementary Material Table S2.

**Acknowledgments:** We would sincerely like to thank those panellists throughout New Zealand who completed our questionnaires. Thanks to Suzie Greenhalgh for her support. Thanks also to our referees for their time, patience, and constructive advice.

**Conflicts of Interest:** The authors declare no conflict of interest. The funders had no role in the design of the study; in the collection, analyses, or interpretation of data; in the writing of the manuscript; or in the decision to publish the results.

## Appendix A

**Table A1.** Standardised parameter estimates from regressions for wearing face masks including number of regional COVID-19 infections.

| | Wore a Face Mask in Public | Wore a Face Mask at Work | While Exercising Outside | Visiting Friends at Their Place | Friends Visiting Your Place |
|---|---|---|---|---|---|
| Involvement with wearing face masks | 0.169 ($p < 0.001$) | 0.126 ($p = 0.009$) | 0.217 ($p < 0.001$) | 0.355 ($p < 0.001$) | 0.338 ($p < 0.001$) |
| Attitude towards face masks | 0.450 ($p < 0.001$) | 0.354 ($p < 0.001$) | | | |
| FC1–Threat of infection is limited | | | 0.176 ($p < 0.001$) | 0.104 ($p < 0.001$) | 0.148 ($p < 0.001$) |
| FM1–Ineffectiveness and impracticality of masks | | | −0.188 ($p < 0.001$) | | |
| Age | −0.077 ($p = 0.004$) | | | 0.190 ($p < 0.001$) | |
| Gender | 0.126 ($p < 0.001$) | 0.126 ($p < 0.001$) | | | −0.094 ($p = 0.006$) |
| Asian | | 0.118 ($p < 0.001$) | 0.153 ($p < 0.001$) | 0.189 ($p < 0.001$) | 0.139 ($p < 0.001$) |
| Income | | | | | −0.119 ($p < 0.001$) |
| Regional case numbers | 0.013 ($p = 0.627$) | 0.002 ($p = 0.947$) | 0.030 ($p = 0.346$) | 0.030 ($p = 0.351$) | 0.034 ($p = 0.328$) |
| Adjusted $R^2$ | 0.35 | 0.24 | 0.21 | 0.24 | 0.22 |
| Observations | 985 | 678 | 816 | 813 | 696 |

**Table A2.** Standardised parameter estimates from regressions for wearing face masks including regional incidence of COVID-19 infections.

| | Wore a Face Mask in Public | Wore a Face Mask at Work | While Exercising Outside | Visiting Friends at Their Place | Friends Visiting Your Place |
|---|---|---|---|---|---|
| Involvement with wearing face masks | 0.170 ($p < 0.001$) | 0.126 ($p = 0.010$) | 0.219 ($p < 0.001$) | 0.356 ($p < 0.001$) | 0.338 ($p < 0.001$) |
| Attitude towards face masks | 0.450 ($p < 0.001$) | 0.355 ($p < 0.001$) | | | |
| FC1–Threat of infection is limited | | | 0.177 ($p < 0.001$) | 0.106 ($p < 0.001$) | 0.151 ($p < 0.001$) |
| FM1–Ineffectiveness and impracticality of masks | | | −0.186 ($p < 0.001$) | | |
| Age | −0.079 ($p = 0.003$) | | | 0.189 ($p < 0.001$) | |
| Gender | 0.125 ($p < 0.001$) | 0.124 ($p < 0.001$) | | | −0.094 ($p = 0.007$) |
| Asian | | 0.122 ($p < 0.001$) | 0.156 ($p < 0.001$) | 0.194 ($p < 0.001$) | 0.143 ($p < 0.001$) |
| Income | | | | | −0.118 ($p = 0.001$) |
| Regional incidence | 0.004 ($p = 0.870$) | −0.021 ($p = 0.545$) | 0.030 ($p = 0.344$) | 0.007 ($p = 0.823$) | 0.022 ($p = 0.530$) |
| Adjusted $R^2$ | 0.35 | 0.24 | 0.21 | 0.24 | 0.22 |
| Observations | 985 | 678 | 816 | 813 | 696 |

**Table A3.** Standardised parameter estimates from regressions for wearing face masks with regional dummy variables.

| | Wore a Face Mask in Public | Wore a Face Mask at Work | While Exercising Outside |
|---|---|---|---|
| Involvement with wearing face masks | 0.157 *** | 0.139 ** | 0.213 *** |
| Attitude towards face masks | 0.464 *** | 0.348 *** | |
| FC1–Threat of infection is limited | | | 0.180 *** |
| FM1–Ineffectiveness and impracticality of masks | | | −0.190 *** |
| Age | −0.088 *** | | |
| Gender | 0.117 *** | 0.119 *** | |
| Asian | | 0.108 ** | 0.150 *** |
| Income | | | |
| Bay of Plenty | −0.017 | 0.017 | −0.017 |
| Canterbury | −0.028 | 0.004 | −0.004 |
| Gisborne | 0.012 | −0.045 | −0.025 |
| Hawke's Bay | −0.024 | 0.057 | −0.009 |
| Manawatu-Whanganui | 0.047 | −0.073 | −0.037 |
| Marlborough | 0.022 | 0.069 | 0.052 |
| Northland | 0.026 | 0.016 | −0.019 |
| Otago | −0.073 ** | 0.004 | 0.019 |
| Southland | 0.001 | 0.009 | −0.044 |
| Taranaki | −0.011 | 0.008 | 0.008 |
| Tasman Nelson | −0.044 | −0.016 | −0.065 * |
| Waikato | 0.048 | −0.040 | −0.034 |
| Wellington | −0.029 | −0.025 | 0.002 |
| West Coast | −0.017 | 0.022 | −0.037 |
| Adjusted $R^2$ | 0.36 | 0.25 | 0.21 |
| Observations | 985 | 678 | 816 |
| | **Visiting Friends at Their Place** | **Friends Visiting Your Place** | |
| Involvement with wearing face masks | 0.355 *** | 0.340 *** | |
| Attitude towards face masks | | | |
| FC1–Threat of infection is limited | 0.106 *** | 0.138 *** | |
| FM1–Ineffectiveness and impracticality of masks | | | |
| Age | 0.188 *** | | |
| Gender | | −0.094 ** | |
| Asian | 0.193 *** | 0.138 *** | |
| Income | | −0.120 *** | |
| Bay of Plenty | 0.003 | −0.025 | |
| Canterbury | −0.017 | −0.071 | |
| Gisborne | 0.009 | 0.001 | |
| Hawke's Bay | −0.041 | 0.009 | |
| Manawatu-Whanganui | 0.015 | 0.046 | |
| Marlborough | −0.068 * | −0.043 | |
| Northland | −0.018 | −0.044 | |
| Otago | −0.035 | 0.005 | |
| Southland | −0.045 | −0.064 | |
| Taranaki | 0.015 | −0.020 | |

**Table A3.** *Cont.*

| | Wore a Face Mask in Public | Wore a Face Mask at Work | While Exercising Outside |
|---|---|---|---|
| Tasman Nelson | | −0.001 | −0.040 |
| Waikato | | −0.016 | 0.031 |
| Wellington | | −0.030 | −0.040 |
| West Coast | | 0.050 | 0.022 |
| Adjusted $R^2$ | | 0.24 | 0.22 |
| Observations | | 813 | 696 |

Notes: Auckland was the reference region in all regressions. * indicates $p < 0.05$, ** indicates $p < 0.01$, *** indicates $p < 0.001$.

**Table A4.** Key correlations.

| | Wore a Mask When You Went Out in Public | Wore a Mask When You Went to Work | Wore a Mask When You Visited Friends at Their Home | Wore a Mask When You Exercised Outside | Wore a Mask When You Had Friends to Visit You at Your Home |
|---|---|---|---|---|---|
| COVID-19 Involvement | 0.419 ** | 0.384 ** | 0.368 ** | 0.335 ** | 0.330 ** |
| COVID-19 Attitude | 0.414 ** | 0.370 ** | 0.341 ** | 0.279 ** | 0.262 ** |
| FC1 | −0.191 ** | −0.182 ** | 0.087 * | 0.117 ** | 0.145 ** |
| FC2 | −0.369 ** | −0.297 ** | −0.298 ** | −0.293 ** | −0.253 ** |
| FC3 | 0.127 ** | 0.126 ** | 0.065 | 0.059 | 0.091 ** |
| Mask Involvement | 0.474 ** | 0.393 ** | 0.425 ** | 0.388 ** | 0.382 ** |
| Mask Attitude | 0.568 ** | 0.466 ** | 0.305 ** | 0.265 ** | 0.235 ** |
| FM1 | 0.297 ** | 0.275 ** | 0.382 ** | 0.356 ** | 0.368 ** |
| FM2 | 0.493 ** | 0.393 ** | 0.175 ** | 0.141 ** | 0.105 ** |

Notes: Values are Pearson correlations, Two-tailed test, * indicates $p < 0.05$, ** indicates $p < 0.01$.

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
