# Peer review of "Motivation, Intention and Opportunity: Wearing Masks and the Spread of COVID-19"

_covid, doi:10.3390/covid3040043_

Round 1

Reviewer 1 Report

1.      Policy word seems illogical in keywords. So it is preferable to add some more keywords in some alphabetical order.

2.      Why authors have chosen New Zealand for investigating the regional differences? Is there any specific reason behind this?

3.      In Abstract section: Provide some achieved digital results for regional differences along with frequency of wearing masks

4.      In introduction provide some the advantages and disadvantages of wearing face masks; so that text can be in line with the paper title.

5.      We all knows “new coronavirus (SARS-CoV-2) identified in late 2019” but some standard and critical evidence must be included like: (i) Automated deep transfer learning-based approach for detection of COVID-19 infection in chest X-rays. (ii) Automatic Diagnosis of Covid-19 Related Pneumonia from CXR and CT-Scan Images

6.      Author have taken Omicron as much serious then why not DELTA which was much dangerous than that of Omicron?

7.      No need to write word see in [see 8, 9, 10, 11].

8.      Line no. 113-115: Give some proof or Govt note for stating “the central government closed New Zealand’s international border, to all except returning citizens and permanent residents, and instituted a four-tier alert system”.

9.      Is there any specific reason for make S1 as red in line no 190.

10.  It would be better if author list some symptoms characteristic of COVID-19 infection somewhere in introduction section.

11.  Some background details about many researchers proposed classification of Covid-19, various types of Pneumonia, Tuberculosis and normal X-ray images must be included to give suitable background to the research. (i) This can be found in Novel deep transfer learning model for COVID-19 patient detection using X-ray chest images (ii) Metaheuristic-based deep COVID-19 screening model from chest X-ray images

12.  How the authors decided that the questionnaire size n=1000 would be enough for this study?

13.  How the gender sensitivity has been taken into consideration? Justify.

14.  Can we consider poor people for wearing mask in this study wrt their financial conditions? Justify.

15.  Line number 465: List the three important implications for recommendations.

16.  Line 487: What is different social settings? Is it different social programs? Which one is proper? Explain with example.

17.  Conclusion section must have some digital values wrt findings of this paper.

18.  As per the title justify how this work provide the opportunity?

19.  Can the author include some mythos about COVID-19 vaccines that they have some side effects? It may support the govt. to successfully enhance the agenda of vaccine also.

20.  Additional References: The following articles could be useful:

-       A novel deep convolutional neural network for diagnosis of skin disease. Traitement du Signal, Vol. 39, No. 5, pp. 1873-1877. https://doi.org/10.18280/ts.390548 

- Efficient automated disease diagnosis using machine learning models

Reviewer 2 Report

This is a well-written manuscript with clearly articulated hypotheses. The questions raised are pertinent, particularly the hypothesis that regional differences in mask use behaviour are not only influenced by beliefs, attitudes and motivations but also by perceived threat.

Major comments

1.       In terms of designing a questionnaire, the definitional distinctions between “attitudes”, “involvement” and “intention” seem intrinsically challenging to delineate and separate. It’s therefore perhaps not too surprising there are strong and statistically significant correlations between these different variables. I think more in-depth explanation around how this challenge was navigated would be helpful. This could be in one or more of the Background, Methods or Discussion sections. More depth could also perhaps be given if necessary in the Supplementary materials.

2.       Assuming “attitudes”, “involvement”, “intention” and “beliefs” have been adequately teased apart and delineated in the questionnaire, correlation between beliefs and other variables tells us nothing about the nature of any causal relationship that exists. In other words, beliefs could influence attitudes / involvement / intention OR attitudes/involvement/intention could influence beliefs. For example, those with strong political views may unwittingly seek out and embrace information / beliefs that support and justify their ideological position. We therefore need to be cautious about inferring from study findings that changing beliefs will in turn influence attitudes / involvement and intention. Moreover, it’s possible that all four variables are similarly influenced by some additional unmeasured variable that hasn’t been captured by the study (eg pre-existing core values / political / ideological beliefs / trust in government etc). Suggest commenting on these potential limitations /caveats in the discussion.

Minor comments

1.       Tables – suggest titles of Tables need to more clearly and explicitly describing what is shown. Using language used in text. For example, Table 2: suggest “Correlation between composite factors and individual belief variables about COVID-19 and COVID-19 vaccinations” would be clearer than current title. It’s currently challenging for the reader to work out which of the four broad variables are being correlated together in each of the Tables. Also suggest labelling both the rows and columns of the table – eg Attitudes versus intention etc. Also need to be clearer around the variables displayed. These are presumably r values, but this should be clearly stated.

2.       In appendiceal material would be good to see plots of key correlations, particularly correlations with actual practice.

Reviewer 3 Report

This paper analyzes survey data regarding attitudes toward COVID-19 mitigation and mask-wearing in New Zealand. It seems rather thorough, with principal component analysis + varimax performed to condense various different attitudes into informative factors, and then a number of linear regressions run. It should be acceptable for publication after substantial revisions.

However, I think it’s very difficult to read and appreciate in its current form. I found myself consistently puzzled while trying to track the details on what was being done at any point. I think the manuscript needs to be carefully rewritten for readability.

Take “Section 3.2 Methods”. (By the way, there are two Section 3’s). This is very hard for me to read because you use too much prose and not enough numbering and notation. You speak of a first part, a second part, and a third part. The “first part” contains both factor analysis (FA) and a linear regression. But then in your results section, you have “Section 3.1 Factor analysis” in which you only describe the FA, not any linear regression. Please rewrite Section 3.2 Methods carefully with subsubsections or bolded paragraph headings, and clearly note that Part 1 will be discussed in Section 4.1, Part 2 will be discussed in 4.2, etc.

In addition, writing out your linear regressions in words gets very confusing. Please use standard notation such as: we perform a linear regression

            Var1 ~ Var2 + Var3 (this on a new line, centered).

Wording like “involvement with slowing the spread of Covid-19 and as the dependent var-237 iable and beliefs about COVID-19 as the independent variables,” is difficult to read. Moreover, it’s sometimes unclear what your predictor variables are. In Table 4, you use the four factors from Table 2 plus the two factors from Table 3, so six predictors in all. But in Table 5, you use just four predictors.

Just to be clear, your labels such as “Threat of infection is limited” are your editorialized description of the factors? I recommend that you make the factor analysis its own subsection (which it currently is in the results but not Section 3.2 Methods). I recommend that you explicitly say something like “we uncover four factors, which we name Threat of infection is limited, Threat to health is exaggerated, Vaccine effects, Modes of infection, and notate as F1, F2, F3 and F4 respectively.” And the same for the two factors in Table 3, maybe G1 and G2 or F5/F6. Then you can specifically write things like “We perform linear regressions

Stopping the spread of COVID-19 saves lives  ~ F1 + F2 + F3 + F4;

Slowing the spread of COVID-19 will keep supply chains functioning ~ F1 + F2 + F3 + F4;

….

Or maybe “We perform linear regressions between five different beliefs and factors F1 to F4

Belief ~ F1 + F2 + F3 + F4

For five different beliefs:….

You could even write these lines twice. You could substitute the more mathematical F1+F2+F3+F4 with a brief phrase like “beliefs on COVID-19” or something more suitable and precise.

 -------------------------------------------------------------------

Sorry, this review is already running on too long and is probably difficult to read itself. I’ll try and summarise my suggestions as carefully as possible.

1) Please explain methodology in much greater detail.

For example, I don’t know what you mean by “PCA followed by varimax *when necessary*”. Aren’t you just performing PCA and then varimax?

For example, I suspect you’ve applied PCA, scaled the unit norm principal components by the length of the eigenvalue, and then applied some cutoff to reduce any small loading to zero, but you should state this, and state the cutoff.

For example, please explicitly state that your labels like “Threat of infection is limited” are your editorialized description of the factors.

For example, in Table 2, what are you performing factor analysis on? Standardised or unstandardized data? Your data are numbers from 1-5 in terms of agreement, is that right?

2) Please structure Section 3.2 Methods so it is as easy as possible to connect subsections of the results section with the parts of your methodology. See the first part of this review before 1).

3) Please make the different parts of your methods and results section easier to track. You should use more mathematical notation in your linear regressions. It is difficult to refer back to Section 3.2 and find a linear regression entirely written out in words. If you write a linear regression like

Belief ~ F1 + F2 + F3 + F4 (4)

then you can give it an equation number in the methods section and then refer back to this equation number in the results section. Alternatively, rewrite the equation in the results section. See the first part of this review before 1).

3a) as an alternative, consider completely restructuring the paper. Rather than all the methods and then all the results, have one top-level section for each of the three parts of your methodology, and each top-level section can contain both methods and results. I would prefer this, as you can state the linear regression you perform and then immediately state the results.

4) Please quote specific numerical values from your tables to prove your points, don’t just refer to the whole table, eg “Beliefs about COVID-19 and COVID-19 316 vaccines explained a substantial proportion of attitudes and involvement regarding slow-317 ing the spread of COVID-19 (see Table 4). Similarly, beliefs about the advantages and dis-318 advantages of wearing face masks explained a substantial proportion of the variance in 319 involvement with, and attitudes towards, wearing face masks (see Table 4).”

5) Please give confidence intervals for your coefficients rather than p-values, which is becoming more standard in statistics. They are more informative.

6) Your supplementary files are missing. In particular, I’d be interested in this: “Involvement scores were computed for each respondent as the simple arithmetic av-228 erage of their agreement ratings for the 10 statements in the involvement scales [8]. Atti-229 tudes scores were computed as the simple arithmetic average of their agreement ratings 230 for the five statements in the attitude scales [8].” What are these 10 and 5 statements? Table 2 has 14 attitude statements.

7) Methodology and mathematics aside, I find the English also difficult to read at parts. There are numerous awkward sentences and typos. Sometimes, the clause structure makes the sentence difficult to read, and could be improved quite easily. Here are some examples below – this is only a brief list. Please edit the paper thoroughly to improve English readability (more readable that this review haha).

“Prior to the emergence of the Omicron variant, we found that, although people’s motiva-9 tions regarding wearing face masks to prevent the spread of COVID-19 in New Zealand were 10 largely similar across regions, there were large differences in the frequency of wearing face masks 11 across the regions.” sentence a bit awkward due to repetition of words and complex clause structure

“Success in translating intentions into actions and changing behaviour 28 also requires that individuals perceive, in a timely fashion, the need to act.” would be simpler to write “perceive the need to act in a timely fashion.”

“For example, governments have advocated wearing face masks with a view to slow-33 ing the spread of COVID-19 thereby avoiding higher rates of infection and mortality and 34 reducing demands on health systems and transport systems and reducing economic [6] 35 and psychological damage [7].” Sentence hard to read, could add comma before thereby and reduce the number of “and”s.

“That is, differences in the per- 46

ceived imminent threat of airborne infection.” not a valid sentence

“Covid-19” or “COVID-19” be consistent

8) There are two Section 3’s, and the references are quite inconsistent, eg with some having a DOI and others none.

9) I think the F-test p-value can be dropped in all the tables. The F-statistic is closely related to the R^2 value, and so the p-value is “always” very small for a linear regression when the R^2 is non-trivial, and is indeed in fact always less than your 0.001 threshold. Perhaps you could just make a comment that the F-test p-value is <0.001 for every linear regression. In fact, I imagine it would be much smaller that 0.001.

I may request some further changes in a next round of revision – I think these changes should make the paper more readable and help me analyze it closer.

Round 2

Reviewer 1 Report

all my concerns have been addressed by the authors

Reviewer 3 Report

Thanks to the authors for their revisions. The authors have done a careful job with their changes (to be honest, they're quite hard to read in the manuscript, but they are well set out in the response letter).

My apologies - I was unaware that supplementary material was accessible via a different link and not included with the manuscript file. My mistake.

I don't think that the introduction, methodology, results format is actually absolutely required - it's usually a guide. In theory, I think the paper could be restructured into three sections, each with methodology and results. If you like, you can restructure the paper that way, if the editor allows. However, I won't make you and I'll suggest acceptance as is.